# Multimodal Diabetes Empowerment for Older Adults with Diabetes

**DOI:** 10.3390/ijerph191811299

**Published:** 2022-09-08

**Authors:** Keumok Park, Youngshin Song

**Affiliations:** 1Department of Nursing, College of Health and Welfare, Woosong University, Daejeon 34606, Korea; 2Department of Nursing, College of Nursing, Chungnam National University, Daejeon 35015, Korea

**Keywords:** diabetes mellitus, older adults, empowerment, self-care, physical function

## Abstract

Systematically improving empowerment is not easy when operating a diabetes program for older adults. This study aimed to develop and test the feasibility of the diabetes empowerment (Dia-Empower) program for older adults with type 2 diabetes. A non-randomized controlled study with a matched sampling design was conducted. Community-dwelling older adults with diabetes were allocated to either the Dia-Empower program group or a control group. Changes in the primary (diabetes self-care and empowerment) and secondary outcomes (body composition and physical function) were compared between the groups. The scores for diabetes self-care and empowerment were significantly higher in the experimental group than in the control group. Changes in skeletal muscle mass and body fat ratio were significantly different between the groups. Handgrip strength and shoulder flexibility positively changed in the experimental group. The Dia-Empower program was feasible for older adults with diabetes in the community. In the future, it is necessary to study the long-term effects of the program and its effects on blood sugar control.

## 1. Introduction

The prevalence and mortality of type 2 diabetes mellitus (T2DM) are increasing worldwide. With the recent spread of the coronavirus worldwide, diabetes reportedly has a higher risk of infection and more than double the severity of the disease compared to the general population, requiring special management [1]. Meanwhile, the burden of diseases that impede socioeconomic development is also increasing [2]. As of 2014, 8.3% of the world’s population, 7.3% of the Korean population, and 22.6% of the Korean population over the age of 65 years had diabetes [3,4]. Individuals with T2DM require a healthy lifestyle through self-management efforts, leading to optimal blood sugar levels [5]. To this end, many studies have successfully applied programs based on various theories to promote diabetes self-management [6,7]. Despite the high prevalence of diabetes in older adults, some studies and active care targets have excluded older adults owing to the presence of multiple comorbidities and high risk [8]. As the American Diabetes Association (ADA) has been providing guidelines for diabetes management to older adults for more than a decade, the need for scientific evidence from diabetes research for older adults is gaining increasing importance.

Several diabetes self-care programs, such as motivational interviewing, coaching, cognitive behavioral therapy, and technology-based interventions, have contributed to the promotion of diabetes self-care and improvements in glycemic control, but not consistently [9]. Application of a single model, the program being expert oriented (led by experts), and ambiguity in the intervention have been noted as possible causes [9]. Patient-oriented multimodal approaches and precise protocols have been recommended as alternatives [9,10].

Empowerment theory, a patient-centered theory, advocates that the role of the healthcare provider is to present the vision, not act as an expert, when educating or running a program [11]. Diabetes empowerment can help develop the decision-maker’s capacity for one to be responsible for one’s own life using a collaborative approach tailored to match the fundamental realities of diabetes care [11,12,13]. However, although some studies have applied education interventions based on empowerment theory [14,15], studies implementing complex intervention strategies, including exercise and education, are lacking [9]. In previous studies, the concept of empowerment was measured only as an outcome variable, without a diabetes empowerment intervention [16,17]. Moreover, the major outcomes of most programs involved psychological factors, such as self-efficacy, and physiological factors, such as weight, waist circumference, and body mass index, have also been reported [9].

Meanwhile, it is well known that physical activity is essential for diabetes self-care and that it improves physical functions. The “Be Happy and Strong” (BeHaS) exercise was developed to improve physical functions (such as grip strength and shoulder flexibility), pain relief, and self-efficacy and to reduce depression in individuals with arthritis, involving exercises that can be easily performed by older adults [18,19]. The BeHaS exercise program has been applied to patients with knee arthritis [18], women with breast cancer [20], older women [21], and patients with hypertension [22]; however, to our knowledge, it has not been applied to older adults with T2DM. Moreover, no studies have applied BeHaS exercise combined with the principles of diabetes empowerment to older adults. Therefore, a multimodal study with the addition of BeHaS exercise for T2DM based on the diabetes empowerment principle is necessary to test its effect on physical function and empowerment in older adults with T2DM. The purpose of this study was to apply the multimodal diabetes empowerment program (Dia-Empower program) and evaluate its effects on diabetes self-care, empowerment, body composition, and physical function in older adults with T2DM. Specifically, we tested the following hypotheses.

First, we hypothesized that a greater change would be noted in the experimental group compared to the control group in the primary outcome (diabetes self-care and empowerment) scores following the intervention. Second, we hypothesized that a greater change would be noted in the experimental group compared to the control group in the secondary outcome (body composition and physical function) scores following the intervention.

## 2. Materials and Methods

### 2.1. Design

A non-randomized control study design was employed to evaluate the effectiveness of the Dia-Empower program on physical function, diabetes self-care, and empowerment in older adults with T2DM.

### 2.2. Sample and Setting

Participants aged 65 years or older with hemoglobin A1c (HbA1c) of 6.5% or higher (diabetic) were recruited among the residents visiting the S City Public Health Center. A recruitment flyer was posted at the public health center. Individuals who voluntarily wished to participate were enrolled in the study after screening for cognitive function. The sample size was calculated using the G*Power program 3.1 version (Heinrich Heine Universität Düsseldorf, Düsseldorf, Germany). A significance level of 0.05, a power (1-β) of 0.80, and an effect size of 0.8 per a previous study were entered into the program [23], and a sample size of 26 was found to be appropriate for each group. Considering the dropout rate (20%), 32 participants were assigned to the experimental group and 31 to the control group using the matching sampling method. We included individuals who could understand the aims of the study with cognitive integrity, were literate, and could perform the program. Those with restricted physical activity or problems performing activities of daily living, those who had participated in other similar exercise programs or had participated in the BeHaS program in the past, and those with cognitive disorders were excluded.

The study was conducted at the Diabetes Education Center of the S-city Public Health Center. To ensure eligibility, participants were screened for HbA1c levels > 6.5% using a portable analyzer in advance. The primary outcomes were surveyed when the criteria for participant selection were met. Primary and secondary data were collected at baseline and at week 8.

A matched sampling design was used to assign 32 participants to the experimental and control groups without randomization or blinding. The participants in the two groups were matched for age, sex, and HbA1c levels. A total of 63 participants were recruited and allocated to either the intervention (*n* = 32) or control (*n* = 31) groups in public health centers. The final number of participants was 31 in the experimental group and 31 in the control group because one participant in the experimental group dropped out due to health problems (falling injury at home). The dropout rate in the experimental group was 3.2%. Compliance with the group sessions for the Dia-Empower program was satisfactory, and no harmful effects were observed during the study.

The recruitment flow for this study is presented in Figure 1.

### 2.3. Primary and Secondary Outcomes

The primary outcomes included diabetes self-care and diabetes empowerment. The secondary outcomes comprised anthropometric measures, such as body weight (BW), body index mass (BMI), skeletal muscle mass (SMM), and body fat ratio (BFR). Physical functions, such as handgrip strength and shoulder flexibility, were also used as secondary outcomes.

Diabetes self-care was measured using the Korean version of the Summary of Diabetes Self-care Activities (SDSCA), and diabetes self-care activities in five areas, including diet, exercise, blood sugar testing, smoking, and foot care, were assessed over the last 7 days. The SDSCA reliability at the time of development was determined at Cronbach’s α = 0.78–0.77. In this study, the total Cronbach’s α for the SDSCA was 0.71.

Diabetes empowerment was measured using the Diabetes Empowerment Scale-Short Form (DES-SF), developed by Anderson et al. In 2003, the Korean version was used to measure diabetes empowerment [24]. The reliability of the DES-SF in this study was the same as that at the time of development, with Cronbach’s α = 0.84.

Anthropometrics included BW, BMI, SMM, and BFR without shoes measured using a stadiometer BSM370^®^ (InBody Co., Seoul, Korea) and body composition analyzer Inbody 770^®^ (InBody Co.) with simultaneous multi-frequency impedance, systolic blood pressure, and diastolic blood pressure measurements with an electronic blood pressure monitor Omron HEM7121^®^ (Omron, Kyoto, Japan). Blood glucose levels were measured with a glucose meter Accu-Chek^®^ (Roche, Indianapolis, IN, USA), and HbA1c was measured using a portable analyzer-type glycated hemoglobin analyzer SD A1cCare ™ (Ethitech, Midrand, South Africa).

The handgrip strength of both hands was measured with an electronic hand dynamometer CAMRY EH101^®^ (CAMRY Scales, Zhongshan, China), and the flexibility of both shoulders was measured using a measuring tape. Shoulder flexibility was measured by placing the patient’s hands behind their back and measuring the distance in cm between the middle fingers using a tape measure.

### 2.4. Process of Dia-Empower Program Development

The Dia-Empower program was developed in four steps:(a)Reviewing the literature and preparing the instructors: Social cognitive and empowerment theory accepted [14,25,26]. Researchers acquired certification through BeHaS instructors and diabetes educators.(b)Designing the program: Program designed for 8 weeks and 8 sessions.(c)Reviewing feasibility and safety based on expert opinions: Feasibility and content validity were reviewed by experts.(d)Evaluating the program: Applied and tested the program.

### 2.5. Intervention (the Dia-Empower Program)

For the intervention group, the Dia-Empower program consisted of diabetes empowerment (10 min), diabetes education (10 min), and exercise (40 min) and was administered by the researcher, Dia-Empower program instructor, and research assistant for 8 weeks (8 sessions), once a week, 60 min per session in class. The content of diabetes education was guided by the Korea Centers for Disease Control and Prevention and the Korean Diabetes Association, and eight topics were instructed and discussed.

The BeHaS exercise was developed based on the Korean traditional “Shimmudo” exercise (low-to-median intensity) (see Appendix A) [18]. This exercise does not require special equipment or facilities for the community, and studies have shown that it can lead to health improvements [18]. Physical exercise consisted of an open mind-body warm-up, warm-up exercise, main exercise, and final exercise to improve muscle strength and joint flexibility that lasted approximately 40 min via a facilitator’s guidance in class; an individual guidebook was also provided. However, no materials were prepared in the empowerment class.

For educational methods applying social cognitive theory, it is necessary to strengthen competence based on the expectation of efficacy and outcome of diabetes self-management. Through small-group (10–11 people) interaction and inclusive education, community-dwelling older adults with diabetes can understand their body, mind, and intrinsic motivation; strengthen their knowledge and competency for disease management; and improve self-management behaviors for diabetes.

To achieve empowerment for diabetes self-care, the “Stages of Behavioral Change in the Diabetes Empowerment Approach” was used to organize the Dia-Empower program [14]. According to Funnell and Anderson’s behavioral change protocol, divided into past–present–future [14], participants used pens and papers to organize their experiences by topic for the conversations and presentations. In the first phase of this Dia-Empower program (1st and 2nd week), the patient develops a degree of self-belief in their change. Participants had time to make autonomous decisions to acquire the knowledge and skills needed to perform self-care. In this stage, the education content consists of diabetes causes, blood sugar management, and its difficulty. The second step (3–5 weeks) was to identify long-term goals for the patient to help the patient choose and commit to action and to support their long-term goals. In the third step, the patient’s efforts were assessed and what the participants had learned in the process was confirmed, through which the empowerment of the patients with diabetes is established. In this phase, participants explored their strengths, discovered resources, and shared their plans. At this point, a sense of responsibility was emphasized. Medication, foot care, and health behaviors were discussed in the education sessions. Finally (6–8 weeks), problem-solving strategies and self-management performance were encouraged, and the participants set long-term goals. There was a process of announcing one’s decision-making and receiving exercise and education certificates. During the educational class for each diabetes management topic, educational materials (one each for 8 weeks) were provided, and the participants shared their experiences with the difficulties in maintaining a healthy diet and with diabetic complications. Table 1 shows the stages and contents of the Dia-Empower program.

For the control group, the usual diabetes education of the public health centers was provided by the staff. The usual diabetes education program, which takes place in class, provides group education to those with diabetes in the community with a basic (2 h) and intensive (2 h) disease education programed by the Korea Centers for Disease Control.

### 2.6. Ethical Consideration

This study was conducted with the approval of the bioethics committee of the university (201711-SB-078-01), and the survey was conducted only for those who voluntarily signed the study participation agreement. If the participants in the control group wanted, an intervention program brochure and a 1-day diabetes camp were provided after the intervention. The participants in the experimental group received compensation of USD 25.

### 2.7. Statistical Analysis

Descriptive statistics, independent t-tests, and the chi-squared test were used to test homogeneity according to the demographics and outcome variables between the groups. The unit of analysis was the group because the data entered into the analysis were the average itself, and no missing data were found. An independent *t*-test was used for statistical differences in outcome variables between the baseline and post-intervention points between groups. Implementation fidelity to this program was evaluated by calculating attrition and assessing the self-reported process log in terms of content, frequency, duration, and intervention dose by a research assistant [26]. The reliability of the SDSCA and DES-SF was calculated using Cronbach’s α. Data were analyzed using the IBM SPSS 24.0 program (IBM Corp., Armonk, NY, USA), and the significance level of all statistical analyses was set at *p* = 0.05, two-sided.

## 3. Results

### 3.1. Homogeneity Test between the Groups

A homogeneity test was performed for the demographics and primary and secondary outcomes, and the results are shown in Table 2. There was no significant difference between the groups in terms of demographics or primary and secondary outcomes.

### 3.2. Hypothesis 1: A Greater Change Would Be Found in the Experimental Group Compared to the Control Group in the Primary Outcome (Diabetes Self-Care and Empowerment) Scores Following the Intervention

The change in the scores of the primary outcomes (diabetes self-care and empowerment) before and after the program was higher in the experimental group than in the control group. Therefore, Hypothesis 1 was supported (Table 3).

The primary outcomes, diabetes self-care and empowerment, were measured using the SDSCA and DES-SF. The changes in the SDSCA and DES-SF scores were significantly different between the groups. The SDSCA scores were significantly higher in the experimental group than in the control group (*t* = −8.16, *p* < 0.001). The total SDSCA score of the experimental group increased by 29.60 (±14.30), whereas that of the control group increased by 3.61 (±10.49). The score changes in the DES-SF were also significantly different between the groups (*t* = −9.04, *p* < 0.001). In the experimental group, the DES-SF score increased by 13.06 (±7.00), while in the control group, it increased by 0.65 (±4.72).

Table 3 presents the changes in the primary outcomes.

### 3.3. Hypothesis 2: A Greater Change Would Be Found in the Experimental Group Compared to the Control Group in the Secondary Outcome (Body Composition and Physical Function) Scores Following the Intervention

The change in the scores of the secondary outcomes (body composition and physical function) before and after the intervention was greater in the experimental group than in the control group but not for all variables. Therefore, Hypothesis 2 was partially supported (Table 4).

The secondary outcomes measured were anthropometric (BW, BMI, SMM, and BFR) and physical functions (handgrip strength and shoulder flexibility on both upper arms).

The results showed that changes in SMM (*t* = 2.32, *p* = 0.024) and BFR (*t* = 3.32, *p* = 0.002) were significantly different between the groups. In the experimental group, SMM increased, while BFR decreased to a greater degree than in the control group. The handgrip strength and shoulder flexibility scores positively changed in the experimental group, but no changes were found in the control group. Table 4 presents the changes in the scores of the secondary outcomes between the groups.

## 4. Discussion

The Dia-Empower program for older adults with T2DM was feasible for diabetes self-care, empowerment, body composition, and physical function. The program focused on strengthening physical exercise and helped improve self-care through diabetes empowerment and education. To achieve the goal of the primary outcomes, such as diabetes empowerment and self-care, diabetes empowerment strategies and education were included for 10 min each before and after exercise in the program. Diabetes empowerment in this program emphasized exploring self-discovery, identifying diabetes problems, making plans, solving problems, expressing future hope, and evaluating goal achievement [10,27]. The empowerment strategy for diabetes self-care was of a question-and-answer format in which the participants spoke of their experience and decision-making process as the center [10,27]. Moreover, the education mainly consisted of practices such as meal making, blood sugar and blood pressure monitoring, and foot-care methods to increase interest and efficiency in this program through empowerment steps. Based on these processes, the levels of diabetes empowerment and self-care were significantly improved in the experimental group.

In contrast, the control group followed the diabetes management program developed by the Korea Centers for Disease Control and participated in regular hospital visits, telephone counseling, and basic and intensive diabetes education in Korea [28]. The diabetes management program applied by the Korea Centers for Disease Control was developed based on the chronic disease model (CDM), which is known as an effective model for managing various chronic diseases, including diabetes. However, a systematic review indicated that CDM does not always improve intermediate- or long-term outcomes in chronic conditions [29]. To achieve a successful CDM outcome for participants with T2DM, the program should be brief, in line with the flow of the participants’ needs, and focus on participant–provider interaction [29]. The Dia-Empower program can be an alternative to complement the weaknesses of interventions based on the CDM.

According to ADA standards for diabetes care in older adults, physical exercise should include aerobic activity, weight-bearing exercise, and/or resistance training for all older adults who can safely engage in such activities [30]. To develop the Dia-Empower program, various diabetes management programs were reviewed, and as a result, the BeHaS exercise was selected. Exercise in the BeHaS program is based on traditional Korean exercise, and its composition includes aerobic activity, balance, and weight-bearing exercises. Moreover, it is easy to follow and promotes intimacy, as it contains components for strengthening relationships (eye contact, hugging, etc.) during exercise [26]. To reflect the characteristics of diabetes, the Dia-Empower exercise program (low to moderate intensity, once a week for 2 months) was less demanding and shorter in terms of intensity and operation time than the conventional program.

After 8 weeks, grip strength and shoulder flexibility in both arms of the experimental group positively changed in this study. Moreover, muscle mass and BFR in the experimental group significantly changed, but body weight and BMI did not. There was no change in body weight and BMI because the frequency of exercise was once a week and the application time was 40 min, which did not follow the ADA recommendation of exercising 2–3 times a week for 150 min. Similar to this study, in a prior study, structured exercise interventions of at least 8-week duration changed HbA1c in people with T2DM without a significant change in BMI [31]. Although previous studies have reported positive physical and emotional changes during exercise [19,21,22,32,33], it is necessary to have a step-by-step goal in consideration of safety when applying exercise. During the program, participants’ safety, such as avoiding falls, hypoglycemia, and orthostatic hypotension, was ensured by placing a chair close to the participants so that they could rest at any time. No adverse events occurred during the study.

Several studies have applied empowerment strategies to chronic diseases, such as chronic renal failure and rheumatoid arthritis, and have succeeded in self-care and symptom management [5,34,35,36,37]. According to ADA guidelines, diabetes self-management education and support should be patient-centered and provided in group or personalized settings [30]. The diabetes empowerment strategy can satisfy ADA guidelines because diabetes empowerment should be operated by considering the needs, goals, and life experiences of patients [5,10]. In Korea, an effective diabetes program should secure access to providers for decision support facilitated through evidence-based guidelines [5,38], but older adults with T2DM in the community still do not participate in preventive healthcare services, such as education, and tailored care services are rarely available [34,39]. In particular, older adults with diabetes who are vulnerable to COVID-19 owing to the spread of infectious diseases are highly likely to reduce their amount of exercise by social distancing and refrain from social activities due to fear of infection, which could result in the deterioration of blood sugar control [1]. The Dia-Empower program, which includes empowerment strategies, education, and exercise, can be used as a safe program that meets all the requirements of an effective program for older adults with diabetes. Furthermore, in previous studies, there was a limitation in evaluating the effect after applying the diabetes self-care program because self-reporting of self-care improvement, such as physical activity, was used. However, this study attempted to overcome this limitation by measuring various body functions to increase the participants’ confidence in the program.

A weakness of this study is that there were no follow-up data after 8-week intervention; thus, it cannot be known how long the effect of the Dia-Empower program lasted. The initial improvement in physiological and behavioral outcomes decreased after approximately 6 months owing to lack of self-management, and knowledge was reduced by approximately 50% [40]. Follow-up after an intervention is important for determining the continuity of program effectiveness. Nevertheless, according to the results of a review study, only approximately 25% of studies reported that post-intervention was provided [40]. As such, this study has some limitations. First, as this program was conducted in the short term, there can be a limit to extrapolating long-term effects, such as HbA1c. We did not monitor HbA1c in this study because HbA1c is measured once every 3 months according to the regulations for the healthcare setting. Second, although matched sampling was used in this study, sampling bias remained. The possibility that this sample bias may have influenced the results cannot be excluded. In the future, it may be necessary to reconfirm the Dia-Empower program through random sampling and blinding. Finally, only trained facilitators, not general healthcare providers, can operate the BeHaS program. Because of its need for expertise, this program can be limited in its dissemination and extension.

However, the low dropout rate (3%) might be a point in favor of the Dia-Empower program compared to the conventional BeHaS program and existing diabetes self-care education programs [22,23]. The reason for the low dropout rate is presumed to be the effect of emphasizing interaction within the group, whereas the BeHaS program composition is tailored to the individual. When specifically assessing the fidelity of other research interventions with respect to the dropout rates, in the exercise session, 31 participants completed the exercise once a week for 8 weeks; however, 25 of them participated in the entire exercise session lasting 40 min, and five participated only in the warm-up and finishing exercises. In the education session, eight education materials were prepared as planned, and 10–12 minutes’ education sessions once a week for 8 weeks were delivered to 29 participants. Two participants attended six of the eight education sessions. Overall, the participation rates were high, and all planned studies were conducted to ensure fidelity. This study found that the Dia-Empower program, including the empowerment strategy, participatory education, and BeHaS exercise, can be applied to older adults with diabetes in the community.

## 5. Conclusions

The Dia-Empower program led to positive results regarding diabetes empowerment, diabetes self-care, and physical functions, such as muscle strength and flexibility, in older adults with diabetes. In practice, the structured and planned Dia-Empower program can be applied to the community and contribute to enhancing diabetes self-care. Additionally, participant-centered education can be disseminated as an educational method for older adults with diabetes. In future studies, it will be necessary to examine whether this program retains its long-term effects using a longitudinal design.

## Figures and Tables

**Figure 1 ijerph-19-11299-f001:**
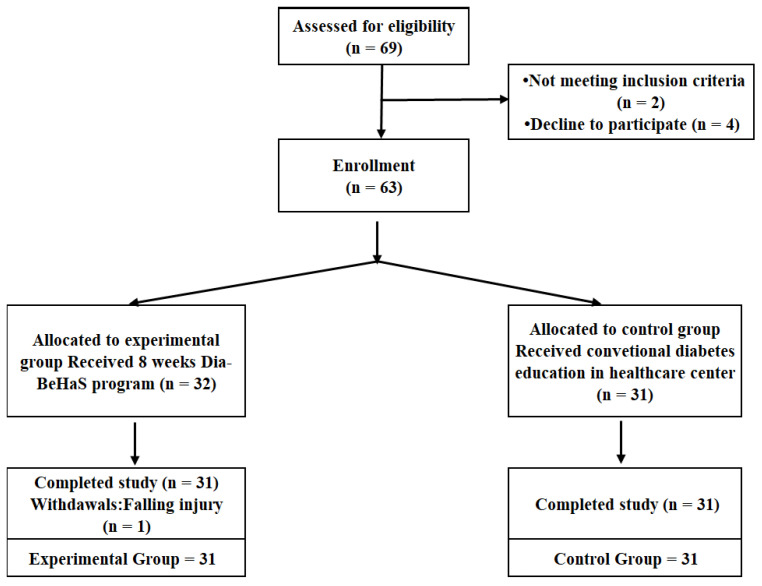
Participant flow diagram.

**Table 1 ijerph-19-11299-t001:** Contents of the Dia-Empower program.

Stages	Week	Empowerment (10 min)	Education (10 min)	Exercise (40 min)	Remarks
Dialog	1st	Understand my body	Overview of diabetes	Body relaxation, Preparation of exercise, Main exercise, (see Appendix A) Final exercise	Pre-test Opening ceremony
	2nd	Who is the owner of my body?	Blood sugar management/Exercise		Practice and group discussion and sharing experiences Diabetes self-care guidebook provided. Practical tasks at home and self-care checklist provided. Capacity building Action task used.
Discovery	3rd	Find my strengths	How to medicate/manage with stress		
	4th	Explain my worth	Foot care		
	5th	My good habits	Smoking/Drinking		
Development	6th	Find my resources	Diabetes diet		
	7th	Become a self-health motivator	Prevention of complications		
	8th	Express hope for the future	Risk notification mark		Post- test & Closing ceremony

**Table 2 ijerph-19-11299-t002:** Homogeneity tests.

Characteristics	Categories	Exp. (*n* = 31)	Cont. (*n* = 31)	*t* or χ²	*p*
		*n* (%) or Mean (±SD)	*n* (%) or Mean (±SD)		
Age	(years)	71.03 (±4.11)	71.45 (±4.76)	0.37	0.712
Sex	Woman	12 (38.7)	16 (51.6)	1.04	0.444
	Man	19 (61.3)	15 (48.4)		
Diabetic Complication	Yes	17 (54.8)	14 (45.2)	0.58	0.446
	No	14 (45.2)	17 (54.8)		
HbA1c	(%)	7.59 (±0.93)	7.25 (±0.61)	−1.75	0.085
Education level	≤Elementary school	13 (41.95)	8 (25.8)	7.48	0.113
	Middle High school	12 (38.7)	16 (51.6)		
	≤College	6 (19.35)	7 (22.6)		
Having Religion	Yes	21 (67.7)	13 (41.9)	4.17	0.073
	No	10 (32.3)	18 (58.1)		
Having a job	Yes	11 (35.5)	9 (29.0)	0.30	0.786
	No	20 (64.5)	22 (71.0)		
Subjective Economic state	Stable	10 (32.2)	11 (35.5)	5.62	0.229
	Moderate	10 (32.3)	15 (48.4)		
	Unstable	11 (35.5)	5 (16.1)		
Alcohol Drinking	No	19 (61.3)	16 (51.7)	9.76	0.534
	Yes	12 (38.7)	15 (48.3)		
Smoking	Yes	3 (9.7)	2 (6.5)	0.22	1.000
	No	28 (90.3)	29 (93.5)		
Duration of DM	(years)	15.35 (±10.91)	13.71 (±9.26)	−0.64	0.524
Cognitive function	(MMSE KC)	27.99 (±1.11)	27.75 (±1.24)	−0.82	0.415
Perceived Health Status	Good	6 (19.3)	7 (22.6)	3.06	0.548
	Fair	11 (35.5)	16 (51.6)		
	Poor	14 (45.2)	8 (25.8)		
Health index	FBS (mg/dL)	145.45 (±40.28)	138.23 (±25.16)	−0.85	0.400
	HbA1c (%)	7.59 (±0.93)	7.25 (±0.61)	−1.75	0.086
	SBP (mmHg)	133.45 (±19.15)	140.58 (±18.93)	1.47	0.146
	DBP (mmHg)	75.68 (±12.61)	79.94 (±9.69)	1.49	0.141
Anthropometrics	Body weight (kg)	63.84 (±9.39)	64.69 (±8.18)	0.38	0.712
	BMI	25.24 (±3.47)	24.71 (±2.70)	−0.66	0.511
	Skeletal muscle mass (kg)	21.75 (±6.22)	24.22 (±4.31)	1.82	0.074
	Body fat rate (%)	34.07 (±8.45)	31.30 (±6.39)	−1.45	0.151
Physical Function	Handgrip strength (Rt.) (kg)	24.17 (±7.00)	24.79 (±5.49)	0.39	0.698
	Handgrip strength (Lt.) (kg)	22.49 (±6.33)	24.04 (±5.93)	0.99	0.327
	Shoulder flexibility (Rt.) (cm)	17.03 (±12.28)	21.39 (±12.13)	1.40	0.165
	Shoulder flexibility (Lt.) (cm)	20.29 (±9.86)	24.16 (±11.45)	1.43	0.159
SDSCA	Total	27.84 (±13.85)	25.90 (±4.97)	0.17	0.864
	Diet	13.94 (±5.82)	12.65 (±5.81)	−0.87	0.386
	Exercise	5.29 (±4.58)	6.35 (±4.79)	0.89	0.375
	Glucose monitoring	4.06 (±4.73)	2.93 (±3.08)	−1.11	0.270
	Foot care	4.55 (±4.23)	6.55 (±4.86)	1.73	0.089
DES	Total	24.68 (±6.71)	25.90 (±4.97)	0.82	0.417

SD, standard deviation; HbA1c, hemoglobin A1c; MMSE KC, Korean version of Mini-Mental State Examination; FBS, fasting blood sugar; DBP, diastolic blood pressure; SBP, systolic blood pressure; BMI, blood pressure; Rt, right; Lt, left; SDSCA, Summary of Diabetes Self-Care Activities; DES, Diabetes Empowerment Scale; Exp., Experiment group; Cont., Control group; DM, Diabetes Mellitus.

**Table 3 ijerph-19-11299-t003:** Changes in the primary outcomes between the groups (Exp. = 31, Cont. = 31).

Variables	Categories	Group	Baseline	Posttest	Changes in Score (Post-Baseline)	
			Mean (±SD)	Mean (±SD)	Diff-Mean (±SD)	*t* (*p*)
SDSCA	Total	Exp.	27.84 (±13.85)	57.45 (±7.71)	29.61 (±14.30)	−8.16 (<0.001)
		Cont.	28.39 (±11.00)	32.00 (±10.25)	3.61 (±10.49)	
	Diet	Exp.	13.94 (±5.82)	22.52 (±3.15)	8.58 (±5.52)	4.48 (<0.001)
		Cont.	12.65 (±5.81)	14.19 (±4.93)	1.55 (±5.92)	
	Exercise	Exp.	5.29 (±4.58)	11.32 (±2.8)	6.03 (±5.39)	4.63 (<0.001)
		Cont.	6.35 (±4.79)	6.77 (±3.48)	0.42 (±4.06)	
	Glucose monitoring	Exp.	4.06 (±4.73)	10.52 (±3.66)	6.45 (±4.65)	6.13 (<0.001)
		Cont.	2.93 (±3.08)	3.26 (±2.72)	0.32 (±3.05)	
	Foot care	Exp.	4.55 (±4.23)	13.09 (±1.72)	8.55 (±4.33)	5.82 (<0.001)
		Cont.	6.55 (±4.86)	7.77 (±4.74)	1.26 (±5.50)	
DES		Exp.	24.68 (±6.71)	37.74 (±2.77)	13.06 (±7.00)	−9.03 (<0.001)
		Cont.	25.90 (±4.97)	25.26 (±4.40)	0.65 (±4.72)	

SD, standard deviation; SDSCA, Summary of Diabetes Self-Care Activities; Exp., Experimental; Cont., Control; DES, Diabetes Empowerment Scale.

**Table 4 ijerph-19-11299-t004:** Changes in the secondary outcomes between the groups (Exp. = 31, Cont. = 31).

Variables	Categories	Group	Baseline	Posttest	Changes in Score (Post-Baseline)	
			Mean (±SD)	Mean (±SD)	Diff-Mean (±SD)	*t* (*p*)
Anthropometrics	Body weight (kg)	Exp.	63.84 (±9.39)	62.94 (±9.41)	−0.90 (±1.59)	0.75 (0.454)
		Cont.	64.69 (±8.18)	64.20 (±7.98)	−0.49 (±1.14)	
	BMI	Exp.	25.24 (±3.47)	24.84 (±3.51)	−0.33 (±0.62)	1.29 (0.201)
		Cont.	24.71 (±2.70)	24.56 (±2.58)	−0.14 (±0.46)	
	Skeletal muscle mass (kg)	Exp.	21.75 (±6.22)	23.64 (±4.94)	1.88 (±5.25)	2.32 (0.024)
		Cont.	24.22 (±4.31)	23.84 (±4.32)	−0.38 (±1.38)	
	Body fat rate (%)	Exp.	34.07 (±8.45)	31.19 (±8.46)	−2.87 (±2.84)	3.32 (0.002)
		Cont.	31.30 (±6.39)	31.21 (±6.52)	−0.08 (±3.70)	
Physical Function	Handgrip strength (Rt.) (kg)	Exp.	24.17 (±7.00)	26.46 (±6.16)	2.29 (±1.96)	−3.96 (<0.001)
		Cont.	24.79 (±5.49)	25.13 (±5.15)	0.34 (±1.93)	
	Handgrip strength (Lt.) (kg)	Exp.	22.49 (±6.33)	24.96 (±6.07)	2.45 (±1.94)	−4.76 (<0.001)
		Cont.	24.04 (±5.93)	24.11 (±5.86)	0.08 (±1.97)	
	Shoulder flexibility (Rt.) (cm)	Exp.	17.03 (±12.28)	11.16 (±10.56)	−5.87 (±5.43)	5.20 (<0.001)
		Cont.	21.39 (±12.13)	21.35 (±11.39)	−0.03 (±3.09)	
	Shoulder flexibility (Lt.) (cm)	Exp.	20.29 (±9.86)	14.87 (±8.06)	−5.42 (±5.71)	4.51 (0.001)
		Cont.	24.16 (±11.45)	23.68 (±11.13)	−0.48 (±2.13)	

SD, standard deviation; Exp., Experimental Group; Cont., Control Group; BMI, Body Mass Index; Rt., right; Lt., left.

## Data Availability

Not applicable.

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
