# Peer review of "Multimodal Diabetes Empowerment for Older Adults with Diabetes"

_ijerph, 2022, doi:10.3390/ijerph191811299_

Round 1
Reviewer 1 Report
I am honored to have reviewed your valuable research results.
The theoretical contribution of this study is that the originality of the research topic stands out in that it can be applied to nursing practice by developing and verifying the Dia-Empower program for the elderly with diabetes.
In compliance with the journal submission regulations, the necessity and purpose of the research are presented in detail and clearly, and it is judged to be faithfully described overall, including logically presenting key concepts and their relevance to existing research.
However, it is expected that this study can be expressed more clearly and logically by supplementing the following several corrections.
Consider replacing “function” written in the keyword with a specific word such as “physical function” and explain it more clearly in the manuscript.
It is recommended to add figures or tables introducing specific interventions for the Dia-Empower program developed in this study, and to supplement the manuscript with readable descriptions.
As the authors described, l am looking forward to conducting and sharing future research on the long-term effects of this Dia-Empower program and its effects on blood-sugar control.
Thank you.
Author Response
Comment 1: Consider replacing “function” written in the keyword with a specific word such as “physical function” and explain it more clearly in the manuscript.
Response: The keyword “function” has been changed to “physical function”
Comment 2: It is recommended to add figures or tables introducing specific interventions for the Dia-Empower program developed in this study, and to supplement the manuscript with readable descriptions.
Response: We have added exercise intervention in the supplement.
Comment 3: As the authors described, l am looking forward to conducting and sharing future research on the long-term effects of this Dia-Empower program and its effects on blood-sugar control.
Response: Thank you for your valuable comment. Based on this pilot study, we wish to conduct a research project and follow the long-term effects.

Reviewer 2 Report
This paper is named: Multimodal Dia-Empower Program for Older Adults with Diabetes. However, as one is reading the abstract it becomes evident that it could be redundant as it basically says, “Diabetes empowerment for people with diabetes”. I suggest removing the “Dia-” to better present the title.
“Function” as a keyword does not provide any particular information regarding the manuscript.
Statistics from the Introduction section should be updated to more recent years, and the contingency from COVID-19 and its effects on people worldwide regarding diabetes should be properly discussed.
Line 87 change .05 to 0.05. this should be corrected throughout the manuscript.
In the Sample and setting section 2.2 authors write the following “those who had participated in other similar exercise programs or had participated in the BeHaS program in the past, and those with cognitive disorders were excluded.” However, they also stated that to their knowledge this program has not been applied to older adults with T2DM (lines 60-63). Please discuss.
Overall, section 2.2 should be improved with Tables and/or diagrams to better follow all parameters that were taken into account for each participant in the study.
Make a list of the four steps of the program at the beginning of section 2.4.
In order to increase the scope of the manuscript, I strongly advise to detailed some of the exercises, for example, the Shimmudo exercise is not know in America, pictures or Figures could greatly improved reading of the paper.
Authors should take into account that this paper, if accepted, could be read by ‘normal people’ affected by the disease, not just scientist or doctors. Hence, they should make efforts to describe the program in a more didactical way. Otherwise, you will greatly reduce the scope of the paper.
‘Wks’ is not a proper simplification of weeks.
Text of sections 3.1 and 3.2 are not properly justified.
At Table 2, is the ‘Education level’ a characteristic that has an impact on whether the person will conclude or not the program?
Following a previous comment, Authors should make efforts to better explained and discussed results from tables 3 and 4.
As it is stated at the conclusions the long-term effects of the Dia-Empower program and its effects on blood-sugar control need to be further investigated. Nonetheless, how feasible is for this goal to be achieved when considering older adults with diabetes?
Moderate English changes required.
Misprints throughout the paper should be carefully revised and corrected.
The materials and methods section should be greatly improved to assure replicability of the Dia-Empower program.
Author Response
Review 2
Comment 1: This paper is named: Multimodal Dia-Empower Program for Older Adults with Diabetes. However, as one is reading the abstract it becomes evident that it could be redundant as it basically says, “Diabetes empowerment for people with diabetes”. I suggest removing the “Dia-” to better present the title.
Response: Thank you for the valuable suggestion. The title has been revised according to the reviewer's suggestion for the reader's understanding, and multimode and diabetes with older adults have been emphasized. Therefore, we changed the title to “Multimodal Diabetes Empowerment for Older Adults with Diabetes”
Comment 2: “Function” as a keyword does not provide any particular information regarding the manuscript.
Response: The keyword “function” has been changed to “physical function”
Comments 3: Statistics from the Introduction section should be updated to more recent years, and the contingency from COVID-19 and its effects on people worldwide regarding diabetes should be properly discussed.
Response: We outlined the risk statistics presented in recent reports on COVID-19 and diabetes and further highlighted this in the discussion section.
In the introduction section “With the recent spread of the coronavirus worldwide, diabetes is reportedly had a higher risk of infection and more than double the severity of disease compared to the general population, requiring special management”
In discussion “In particular, older adults with diabetes who are vulnerable to COVID-19 owing to the spread of infectious diseases are highly likely to reduce their amount of exercise by social distancing and refrain from social activities owing to fear of infection, which could result in the deterioration of blood sugar control.
Comments 4: Line 87 change .05 to 0.05. this should be corrected throughout the manuscript.
Response: "0"s have been placed before the decimal point throughout the manuscript.
Comments 5: In the Sample and setting section 2.2 authors write the following “those who had participated in other similar exercise programs or had participated in the BeHaS program in the past, and those with cognitive disorders were excluded.” However, they also stated that to their knowledge this program has not been applied to older adults with T2DM (lines 60-63). Please discuss.
Response: Although the Behas program has never been applied to diabetes, but it has been applied to arthritis or hypertension, in this study, older adults who had participated in previous similar programs (BeHaS_ were excluded from the participants in this study.
Comments 5: Overall, section 2.2 should be improved with Tables and/or diagrams to better follow all parameters that were taken into account for each participant in the study.
Response: The recruitment flowchart for this study has been presented in figure results according to the reporting guideline (CONSORT).
Comment 6: Make a list of the four steps of the program at the beginning of section 2.4.
Response: We considered making a list; however, there were many tables. Thus, we briefly modified it as follows to improve readability.
The Dia-Empower program was developed in four steps:
- Reviewing the literature and preparing the instructors: Social cognitive and empowerment theory accepted [14,25,26]. Researchers acquired the certification through BeHaS instructors and diabetes educators.
- Designing the program: Program designed for 8 weeks and 8 sessions.
- Reviewing the feasibility and safety based on expert opinions: Feasibility and content validity were reviewed by experts.
- Evaluating the program: Applied and tested the program.
Comment 7: In order to increase the scope of the manuscript, I strongly advise to detailed some of the exercises, for example, the Shimmudo exercise is not know in America, pictures or Figures could greatly improve reading of the paper.
Response: As accurately mentioned by the reviewer, many reader would be unfamiliar with traditional movements such as “Shimmudo.” Therefore, the pictures of the exercise (sample) used in this study are presented in the supplement 1.
Figure S1: The pictures of BeHaS exercise of this Dia-Empower program
Comment 8: Authors should take into account that this paper, if accepted, could be read by ‘normal people’ affected by the disease, not just scientist or doctors. Hence, they should make efforts to describe the program in a more didactical way. Otherwise, you will greatly reduce the scope of the paper.
Response: Although, the text may seem cumbersome for all readers, especially those from a non-medical or non-scientific background, we aimed to present the information as an academic contribution. Elaborate explanation of the procedures could dilute the impact of the paper making it unnecessary lengthy. However, if the reviewer wishes that we incorporate certain specific additional information, we are willingly to definitely attempt the incorporate the respective recommendations to increase the scope of the paper.
Comment 9: ‘Wks’ is not a proper simplification of weeks.
Response: Wks has been revised to “Week” in Table 1.
Comment 10: Text of sections 3.1 and 3.2 are not properly justified.
Response: Since result 1 was not for the purpose of the study, it has been moved to the discussion section concerning the appraisal of the program. The complicated words in Table 1 have been replaced with simple words. However, the homogeneity test at the baseline between the two groups should be presented in the study results according to reporting guideline (CONSORT).
Discussion: 3.1 moved to discussion
“However, the low dropout rate (3%) might be a point in favor of the Dia-Empower program compared to the conventional BeHaS program and existing diabetes self-care education programs [22,23]. The reason for the low dropout rate is presumed to be the effect of emphasizing interaction within the group, whereas the BeHaS program com-position is tailored to the individual. When specifically assessing the fidelity of other research interventions with respect to the dropout rates, in the exercise session, 31 participants completed the exercise once a week for 8 weeks; however, 25 of them participated in the entire exercise session lasting 40 minutes and five participated only in the warm-up and finishing exercises. In the education session, eight education materials were prepared as planned, and 10–12 minutes’ education sessions once a week for 8 weeks were delivered to 29 participants. Two participants attended six of the eight education sessions. Overall, the participation rates were high and all planned studies were conducted to ensure fidelity”
Comment 11: At Table 2, is the ‘Education level’ a characteristic that has an impact on whether the person will conclude or not the program? Following a previous comment, Authors should make efforts to better explained and discussed results from tables 3 and 4.
Response: The reason for evaluating the homogeneity of the education level was that since this program had content to increase the knowledge and understanding for self-management of diabetes, it was necessary to identify the homogeneity between the two groups on an education level.
Comment 12: As it is stated at the conclusions the long-term effects of the Dia-Empower program and its effects on blood-sugar control need to be further investigated. Nonetheless, how feasible is for this goal to be achieved when considering older adults with diabetes?
Response: Behavioral therapy alone cannot determine the factors controlling blood sugar since several factors may play a role. However, the dropout rate of the elderly who participated in this program was low and that they had sufficient interest in and motivation for self-management.
Comment 13: Moderate English changes required. Misprints throughout the paper should be carefully revised and corrected. The materials and methods section should be greatly improved to assure replicability of the Dia-Empower program.
Response: We have re-edited it the manuscript through a professional English editor. The methods section 2.4 was briefly shortened to reduce repetition of content in the material and method section.

Round 2
Reviewer 2 Report
Authors have addressed all my concerns.
I am aware many journals of MDPI asks for 'graphical abstracts' I strongly recommend to authors to include one, representing their four steps of the Dia-Empower program and a sketch of the Shimmudo exercises to further attract the attention of readers.